# Windblown Sand-Induced Degradation of Glass Panels in Curtain Walls

**DOI:** 10.3390/ma14030607

**Published:** 2021-01-28

**Authors:** Yuxi Zhao, Rongcheng Liu, Fan Yan, Dawei Zhang, Junjin Liu

**Affiliations:** 1College of Civil Engineering and Architecture, Zhejiang University, Hangzhou 310058, China; yxzhao@zju.edu.cn (Y.Z.); 21612175@zju.edu.cn (F.Y.); dwzhang@zju.edu.cn (D.Z.); 2China Academy of Building Research, Beijing 100000, China; liujunjin@cabrtech.com

**Keywords:** degradation of glass panels, effective area ratio, relative mass loss, visible light transmittance, windblown sand

## Abstract

The windblown sand-induced degradation of glass panels influences the serviceability and safety of these panels. In this study, the degradation of glass panels subject to windblown sand with different impact velocities and impact angles was studied based on a sandblasting test simulating a sandstorm. After the glass panels were degraded by windblown sand, the surface morphology of the damaged glass panels was observed using scanning electron microscopy, and three damage modes were found: a cutting mode, smash mode, and plastic deformation mode. The mass loss, visible light transmittance, and effective area ratio values of the glass samples were then measured to evaluate the effects of the windblown sand on the panels. The results indicate that, at high abrasive feed rates, the relative mass loss of the glass samples decreases initially and then remains steady with increases in impact time, whereas it increases first and then decreases with an increase in impact angle such as that for ductile materials. Both visible light transmittance and effective area ratio decrease with increases in the impact time and velocities. There exists a positive linear relationship between the visible light transmittance and effective area ratio.

## 1. Introduction

Glass panels are widely used, play an important role in curtain walls, and directly affect the performance of curtain wall structures. One cause of glass panel degradation is windblown sand. As they are exposed to windblown sand, glass panels are affected continuously by grit. Slight damage (such as hollows and scratches) may occur and gradually accumulate, affecting the safety and serviceability of the panels. The deterioration behaviors of glass panels are of significant interest to engineers and researchers in the context of curtain wall structures, e.g., for the reasonable maintenance and timely repair or for the replacement of glass panels and prolongation of the service time of glass curtain walls.

There have been many studies on the degradation behaviors of building materials owing to the effects of windblown sand. Li [1] measured the parameters for evaluating the windblown sand resistance of concrete using an airflow sand-carrying jet test method. Wang et al. [2,3] and Ju [4] studied the windblown sand resistance of cement paste, mortar, and concrete under different impact wind velocities and impact angles using the same test method and determined the mass loss rate and period when entering the steady erosion stage. Jiang [5] and Guo [6] studied the variation rule of the relative mass loss, impact angles, and impact velocities of cement mortar under the effect of salt soaking erosion and multiple factors of freeze–thaw and dry–wet cycles in a compound salt solution. Zhang [7] modified the epoxy resin with the addition of SiC micro-powder or both SiC micro-powder and polyurethane toughener and evaluated the mechanical properties and erosion wear properties of the modified epoxy resin. Based on the Taguchi method, Hao et al. [8] found that the impact angle had the largest impact on concrete erosion, which reached 72.48%. Hao et al. [9], Li [10], and Zhang et al. [11,12] used the same method to simulate windblown sand and analyzed the surface morphology by scanning electron microscopy (SEM), thereby studying the degradation behaviors and mechanisms of the surface coatings on steel structures and nano-titanium dioxide films on glass surfaces, respectively. There are many experimental studies on the erosion wear of new materials, e.g., glass fiber/unsaturated polyester composites [13], WC-Co cemented carbide [14], glass-ceramic composite coating [15], fluoroplastic-steel composite tubes [16], carbon fiber reinforced polymer [17], and glass fiber reinforced plastics [18].

The windblown sand that degrades glass materials has also been studied. Li [19] conducted an experimental study and finite element simulations of the windblown sand erosion of toughened glass under temperature and humidity fluctuations and ultraviolet rain. Fan et al. [20] studied the erosion mechanism, surface pattern, and process performance of micro-abrasive water jets on brittle glass. Hao et al. [21] designed the experiment of toughened glass based on the Taguchi method. It was found that the degree of effect on the relative mass loss of toughened glass was mainly due to the impact angle, the impact velocity, and the abrasive feed rate. Zhao [22] and Hao et al. [23] studied the damage mechanism of tempered glass by windblown sand erosion under the combined effect of the freeze–thaw cycle and ultraviolet radiation. Bouzid and Azari [24], and Ismail et al. [25] explored mechanisms of glass degradation. They pointed out that erosion pits caused by mass loss are formed when transverse cracks of the subsurface stratum develop along the direction parallel to the surface until they intersect the surface; radial cracks are formed when cracks on the subsurface stratum develop upward along the vertical direction of the surfaces, ultimately intersecting with the surface (as shown in Figure 1). Ismail et al. [25] also found an exponential function relationship between the sizes of the damage hollows and impact velocities; the exponents were between 1.65 and 2.

The degradation of glass panels under a gas–solid two-phase flow was explored by Hao et al. [26] using the airflow sand-carrying jet test method. The results showed that there were many damage modes in the damage process of glass panels by Scanning Electron Microscopy (SEM). Then, Hao et al. [26] pointed out that the meso-damage modes of engineering glass have two types: a combination of low-angle micro-cutting and high-angle elastic-plastic deformation or a combination of low-angle micro-cutting and damage from high-angle fatigue crack propagation.

In summary, the test methods for windblown sand damage are relatively mature and have been introduced into the research field of glass panel damage. However, since the existing studies on the windblown sand damage of glass panels have focused on the degradation mechanism, there have been few studies on the degradation behavior (s) of glass panels. Therefore, in this study, the windblown sand-induced degradation of glass panels was studied using the airflow sand-carrying jet test method. Focusing on the safety and serviceability changes in glass panels, our experiments explored the long-term performance degradation of glass panels caused by windblown sand. Based on the value obtained, the tendencies of the relative mass loss, the visible light transmittance, and effective area ratio have been obtained, which predict the state of glass panels under service conditions.

## 2. Experimental Program

### 2.1. Materials

The test material was a 6 mm thick piece of float glass; float glass is widely used in curtain walls, owing to its good smoothness and transparency. The basic information on the material is shown in Table 1, based on the initial state of the glass panels.

### 2.2. Specimens

The specimens were 50 mm × 50 mm float glass panels with 6 mm thickness, as listed in Table 2. The specimens of the P_a_ group were used for studying the mass loss of the glass panels, whereas the specimens of the P_s_ group were used for studying the visible light transmittance and effective area ratio.

### 2.3. Experimental Equipment

A sandblasting machine was applied to simulate windblown sand damage. An air compressor was used as the air power source, as schematically illustrated in Figure 2. The high-speed air provided by the air compressor flowed through the pipe connected to the sandbox, causing local low pressure. Then, sand in the sandbox was sucked into the pipeline owing to the pressure difference, forming the sand flow. The gas–solid two-phase flow mixed with the high-speed flow and the sand flow impinged on the samples.

In the testing box, the specimens were clamped at different heights and angles using a specimen bracket, as shown in Figure 3. The angles and distances of the specimens were adjustable. After the specimens were clamped, the angles (i.e., the acute angles between the upper surface of each specimen and central axis of the corresponding nozzle) were adjusted to the target impact angles listed in Table 2. The distance between the center of the upper surface of each specimen and the nozzle was adjusted to 20 cm.

### 2.4. Testing Methods

The specimens were clamped on the brackets, as shown in Figure 3, and then, they were adjusted to the target heights and impact angles. To keep the air–sand phase flow stable, it was necessary to keep the sandblasting machine on for two minutes before eroding the specimens. After the impact time of each group of P_a_ and P_s_ specimens reached the target time in Table 2, the specimens were removed from the specimen bracket.

#### 2.4.1. Mass Loss

An electronic balance was used to weigh the remaining mass of the P_a_ specimens. The mass was weighed after blowing off the remaining sand and debris on the surface to ensure that the weighed mass did not contain the masses of such sand and debris.

#### 2.4.2. Visible Light Transmittance

After weighing the mass, the surfaces of the P_s_ specimens were cleaned with water and an acetone solution. Then, the spectral transmittance of the P_s_ specimens was measured by the visible light transmittance ratio and shading coefficient detector.

#### 2.4.3. Meso-Morphology

The surface morphology of the P_s_ specimens was observed via TM3000 desktop SEM (Hitachi, Japan), and photographs were taken. The photographs were then processed using an image processing software (Image J, version 1.48, developed by National Institutes of Health) to extract the damaged area. The percentage of damaged areas with respect to the total area was calculated using the software.

## 3. Results and Discussion

### 3.1. Damage Mode

The mechanism of glass panel degradation is characterized by accumulation of the meso-damage of the glass panel. Hao et al. [26] conducted an experimental study on windblown sand damage of glass panels, pointing out that the meso-damage modes of engineering glass have two types: a combination of low-angle micro-cutting and high-angle elastic-plastic deformation or a combination of low-angle micro-cutting and damage from high-angle fatigue crack propagation. In our study, these damage modes can be observed on the surface of the damaged specimens using SEM, as shown in Figure 4, and are similar to those described by Hao et al. [26]. Ismail et al. [25] and Li [27] introduced these meso-damage modes and their generation mechanisms. Based on these studies, the damage modes in Figure 4 can be classified into three modes: cutting, smashing, and plastic deformation. **Cutting** is the removal of surface material caused by the tangential forces of the impact particles. **Smashing** is material breaking caused by the normal force of the impact particles, as transverse microcracks in the subsurface layer propagate to the surface. **Plastic deformation** is the plastic extrusion deformation caused by the normal force of the impact particles.

These three damage modes coexist in the erosion process and are affected by the impact angle and impact velocity. When the impact angles are small, the tangential force of the impact particles is dominant and the cutting mode develops well. When the impact angles are large, the normal force plays a larger role, and smashing and plastic deformation become the main failure modes. In addition, with the growth of the wind force, the impact velocity of the particles increases and the plastic deformation develops better than the smashing mode (brittle damage), owing to the increase in strain rate at high-speed impact [27].

### 3.2. Relative Mass Loss

The degree of erosion impact is evaluated based on the relative mass loss. The relative mass loss of the specimens is related to the impact time, impact angles, and impact velocities and is defined as follows:(1)$$E=\frac{\Delta M}{mT}$$
where E represents the relative mass loss (mg/g), Δ*M* is the mass loss of the specimens after impact (mg), *m* is the abrasive feed rate (g/s), and *T* is the impact time (s).

#### 3.2.1. Effect of Impact Time on Relative Mass Loss of Glass Specimens

By subtracting the mass measured at each time from that measured at the initial time (impact time of 0 s), the mass loss Δ*M* of the P_a_ specimens at each impact time *t* can be obtained. Then, according to Equation (1), the relative mass loss is calculated at different impact velocities and impact angles with the increase in impact time, as shown in Figure 5.

From Figure 5, the relative mass loss of the specimens can be divided into two stages: decreasing stage and steady stage. Nearly all curves have a tendency to proceed in two stages, except for the group at 17.4 m/s and 90°. This can be explained as follows: at the beginning, the surfaces of the glass specimens are smooth, meaning that cutting (at low impact angles) and smashing/plastic deformation (at high impact angles) develop well. Therefore, the value of the relative mass loss is high. Then, with the specimens degraded, the surfaces gradually become rough. Rough surfaces are detrimental to the development of cutting, causing a decrease in the relative mass loss at low impact angles. As for the specimens at high impact angles, the rough surfaces caused by the sand of the specimens appear as many erosion pits, which cause the particles to bounce off in all directions and to collide with the incident particles, reducing the impact velocities of incident particles. Therefore, it causes a reduction in the relative mass loss of the specimens. Therefore, the relative mass loss of the specimens at high impact angles gradually decreases. After degradation for a long time, the rough surfaces remain stable, corresponding to the steady stage.

It can also be seen from Figure 5 that the relative mass loss at 26.34 m/s is higher than that at 17.4 m/s, which should be evident: degradation at a high kinetic energy is more significant than that at low kinetic energy. Thus, the kinetic energy of sand particles at 26.34 m/s is higher than that at 17.4 m/s, causing a higher relative mass loss.

#### 3.2.2. Effect of Impact Angles on Relative Mass Loss of Glass Specimens

There are two main types of materials with respect to fracture damage modes: ductile and brittle. According to Sheldon and Finnie [28], the ductile and brittle mode curves for the erosion (relative mass loss) with an increase in the impact angle are shown in Figure 6. The relative mass loss of a ductile material includes two stages: in the initial stage, the relative mass loss increases with an increase in impact angle. In the second stage, the relative mass loss decreases with an increase in the impact angle. Therefore, the maximum relative mass loss of the ductile materials appears at an impact angle of 20° to 30°. In contrast to ductile materials, the relative mass loss of brittle materials increases with an increase in the impact angle, and the maximum relative mass loss appears at an impact angle of 90°.

The relative mass losses at different impact velocities and impact times with the increase in impact angle are shown in Figure 7. It can be seen from Figure 7 that the relative mass loss initially increases with an increase in the impact angle and then decreases.

Despite being a typical brittle material, the tendency of the float glass in Figure 7 is not in accordance with the curve of typical brittle materials (shown in Figure 6). This can be explained as follows. As compared with 110 g/min (1.83 g/s) in the reference of Hao et al. [29], the abrasive feed rates in this experiment are very high, i.e., 15.5 g/s at an impact velocity of 17.4 m/s and 15.0 g/s at 26.34 m/s. According to Section 3.2.1, the rough surfaces (caused by the sand) of the specimens exhibit many erosion pits, limiting the reversal of sand particles and reducing the mass loss of the specimens at high impact angles. The higher the impact angles, the lower the mass loss. Therefore, the relative mass loss at 90° is much lower than that at 60°.

### 3.3. Visible Light Transmittance

The full spectral transmittance *τ* (*λ*) of the P_s_ specimens was measured by a visible light transmittance ratio and shading coefficient detector, as shown in Figure 8a. The visible spectral transmittance *τ* (*λ*) (wavelength from 380 nm to 780 nm) of the P_s_ specimens was intercepted from the full spectral transmittance *τ* (*λ*) for analysis, as shown in Figure 8b. Based on this, the visible light transmittance *τ_v_* for each specimen was calculated in accordance with the specification in GB/T 2680-1994 [30].

The visible light transmittance values are obtained for the P_s_ specimens, as shown in Figure 9. It can be seen from Figure 9 that the visible light transmittance *τ_v_* of the specimens decreases gradually with an increase in the impact time owing to the damage accumulation. In addition, the visible transmittance *τ_v_* decreases with an increase in the impact velocity *v*. This is the result of the increase in erosion damage strength. As known, a higher velocity means higher destructive power, causing the decrease in visible light transmittance.

### 3.4. Effective Area Ratio

After the damage area is extracted by the Image J software, the percentage of damaged areas to the total area is calculated using the software. The value after deducting the proportion of damaged areas from the total area (100%) is defined as the effective area ratio *R_AE_* (%).

The effective area ratio was obtained with the image J software, as shown in Figure 10. From Figure 10, it can be seen that the effective area ratio *R_AE_* decreases with the increase in impact time, as expected; *R_AE_* also decreases with increases in the impact angle *α* and impact velocity *v*.

The decrease in effective area ratio *R_AE_* with the increase in impact velocity *v* is the result of the increase in erosion damage strength. The decrease in effective area ratio *R_AE_* with the increase in impact angle *α* may be owing to the fact that, when the normal direction component of the impact particles is large, additional cracks will appear on the subsurface stratum, promoting further development of cracks. It may make more areas of the surface vulnerable to damage. With an increase in the impact angle *α*, the normal direction component of the impact particles increases; thus, smashing and plastic deformation develop better, resulting in a decrease in the effective area ratio.

Assuming that the effective area ratio *R_AE_* of the panel is 100% without damage, then each *R_AE_-t* curve passes through the point (0, 100). By fitting the experimental data in Figure 8 with data processing software (Origin 8.0, developed by OriginLab and 1stOPT8.0, developed by 7D-Soft High Technology Inc.), an equation for the effective area ratio and time can be obtained, as follows:(2)$$R_{AE}=\frac{1}{0.01+K_{RAE}t}$$
where *K_RAE_* is a parameter related to the impact angle *α* and impact velocity *v* and is defined as the effective area erosion coefficient. Based on the test data, the values of *K_RAE_* are obtained by fitting, as follows:$$K_{RAE}=\left\{\begin{array}{c} 3.04\times 10^{-4}\;v=17.40\;m/s\;\alpha =30^{\circ }\\ 4.21\times 10^{-4}\;v=17.40\;m/s\;\alpha =60^{\circ }\\ 4.91\times 10^{-4}\;v=17.40\;m/s\;\alpha =90^{\circ }\\ 5.98\times 10^{-4}\;v=26.34\;m/s\;\alpha =30^{\circ }\\ 6.87\times 10^{-4}\;v=26.34\;m/s\;\alpha =60^{\circ }\\ 7.79\times 10^{-4}\;v=26.34\;m/s\;\alpha =90^{\circ }\\ 8.63\times 10^{-4}\;v=35.28\;m/s\;\alpha =30^{\circ }\\ 1.02\times 10^{-4}\;v=35.28\;m/s\;\alpha =60^{\circ }\\ 1.07\times 10^{-4}\;v=35.28\;m/s\;\alpha =90^{\circ } \end{array}\right.$$

According to the above values of *K_RAE_*, an equation can be obtained by data processing software (1st OPT) fitting, as follows:(3)$$K_{RAE}=-1.169\times 10^{-4}+7.616\times 10^{-6}vlnv+2.328\times 10^{-4}\sin^{2.5}\alpha$$
where sin α reflects the erosion effect of certain physical quantities in the normal direction of the specimens and “*vlnv*” reflects the relationship between the erosion effect and erosion impact velocity *v*.

### 3.5. Relationship between Visible Transmittance and Effective Area Ratio

It is reasonable that, with an increase in the damaged area, the visible transmittance *τ_v_* will decrease. Therefore, the visible transmittance *τ_v_* is positively correlated with the effective area ratio *R_AE_*. The quantitative relationship between the visible transmittance *τ_v_* and effective area ratio *R_AE_* is discussed below.

Figure 11 plots the tested data of the visible transmittance *τ_v_* and effective area ratio *R_AE_* for each specimen. As shown, an approximately positive linear relationship exists between *τ_v_* and *R_AE_*.

To obtain their quantitative relationship, the analysis needs to be extended to a non-damaged condition of the glass specimen. For a non-damaged specimen, the effective area ratio is defined as 100% whereas the visible light transmittance is 92.34%, as shown in Table 1. However, if the impact time is long enough, the entire surface of the glass panel will be damaged and the effective area ratio can be regarded as 0; however, the panel still transmits light. This means that the relationship between the visible light transmittance and effective area ratio is different between low and high effective area ratios. Therefore, it is assumed that, when the effective area ratio is 0, the visible light transmittance of the specimen has a threshold. This threshold needs to be studied in the future.

Based on the above boundary conditions, a simplified equation for the visible light transmittance *τ_v_* and effective area ratio *R_AE_* is obtained using data processing software, as follows:(4)$$\tau_{v}\left(R_{AE}\right)=-110.902+2.085R_{AE}\;\left(54.419\leq R_{AE}\leq 97.431,\;R^{2}=0.872\right)$$
where the value of the effective area ratio *R_AE_* ranges from 54.419 to 100. The value of 54.419 is determined based on the minimum visible light transmittance. The value of 97.431 is determined based on the maximum visible light transmittance.

According to Equation (4), for every 1% decrease in the effective area ratio *R_AE_*, the visible light transmission ratio *τ_v_* decreases by 2.085% when the effective area ratio ranges from 54.419 to 97.431.

## 4. Conclusions

Based on a simulation of a sandstorm using a gas–solid two-phase flow, a windblown sand experiment was conducted to explore the degradation law (s) of glass panels. The following conclusions can be drawn:(1)There are three damage modes in the glass panels subject to windblown sand: cutting, smashing, and plastic deformation. At low impact angles, the cutting mode predominates, whereas under high impact angles, the smashing and plastic deformation modes are dominant. In addition, with the growth of the wind force, the impact velocity of the particles increases and the plastic deformation develops better than the smashing mode (brittle damage), owing to the increase in strain rate at high-speed impact.(2)With an increase in the impact time, the relative mass loss initially decreases and then remains steady. In contrast, with an increase in the impact angle, the relative mass loss initially increases and then decreases, which exhibits the properties of ductile materials.(3)With increases in the time or impact velocity, the visible light transmittance decreases gradually owing to damage accumulation, as expected.(4)The tendency of the variation in the effective area ratio under different situations is similar to that of the visible light transmittance. There exists an approximately positive linear relationship between the visible light transmittance and effective area ratio.

In summary, the experiment simulated the degradation of glass panels subjected to windblown sand. We can get the tendencies of the relative mass loss, the visible light transmittance, and effective area ratio, which are similar to the degradation of glass panels under service condition. However, further study is needed on damage accumulation to predict the service life of glass panels.

## Figures and Tables

**Figure 1 materials-14-00607-f001:**
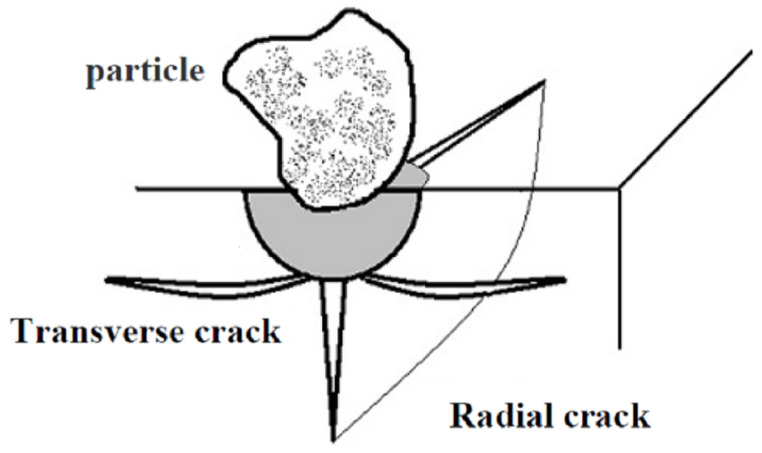
Erosion damage mechanisms of glass [25].

**Figure 2 materials-14-00607-f002:**
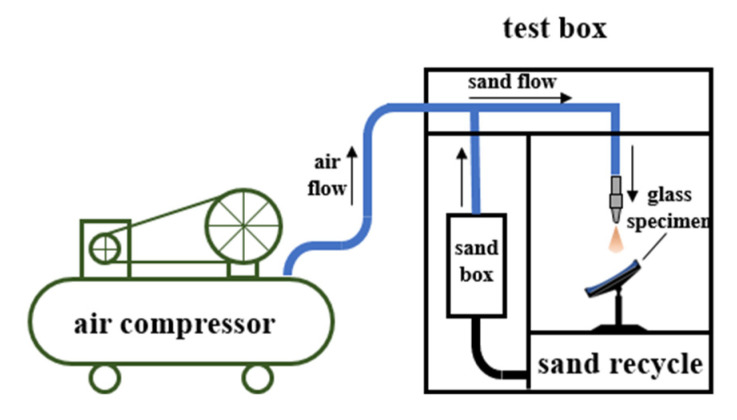
Schematic drawing of the sandblasting equipment.

**Figure 3 materials-14-00607-f003:**
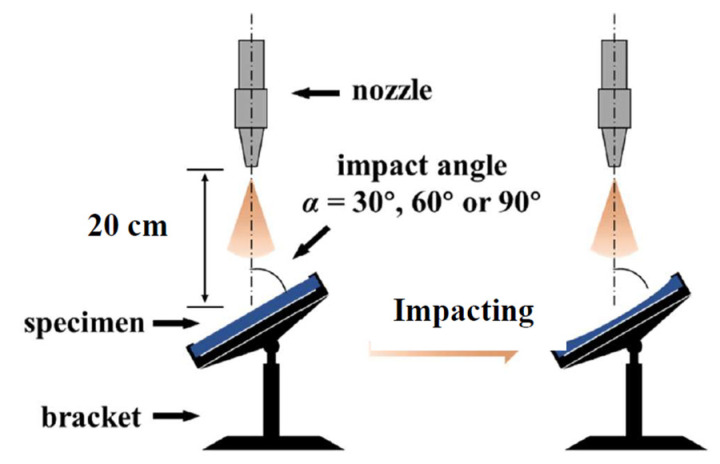
Location of the glass sample.

**Figure 4 materials-14-00607-f004:**
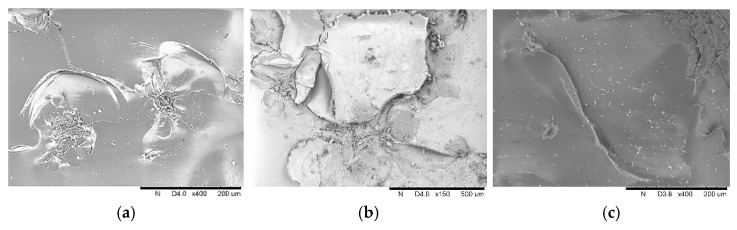
Damage modes of glass specimens: (**a**) cutting, Ps-10-30; (**b**) smashing, Ps-10-90; and (**c**) plastic deformation, Ps-12-30.

**Figure 5 materials-14-00607-f005:**
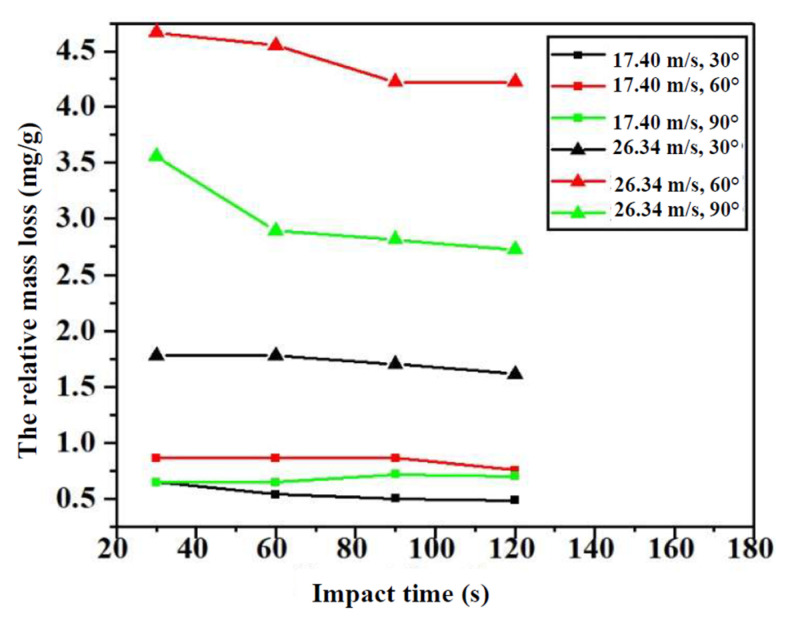
Relative mass loss of glass specimens with an increase in impact time.

**Figure 6 materials-14-00607-f006:**
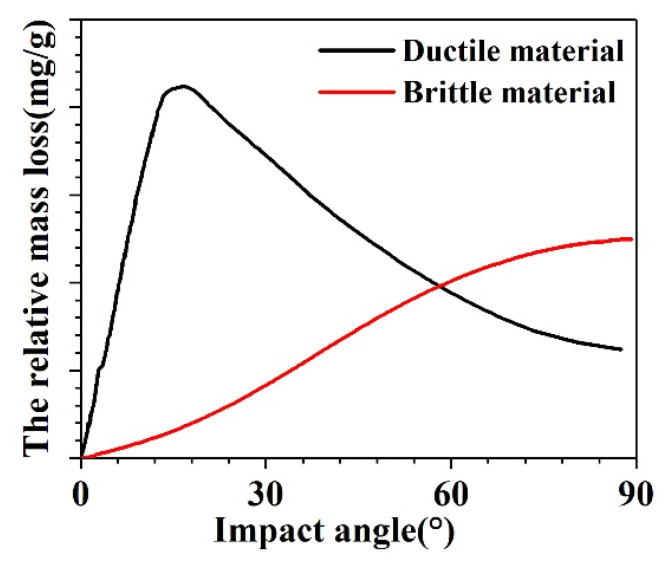
Relative mass loss of ductile and brittle materials with an increase in impact angle [28].

**Figure 7 materials-14-00607-f007:**
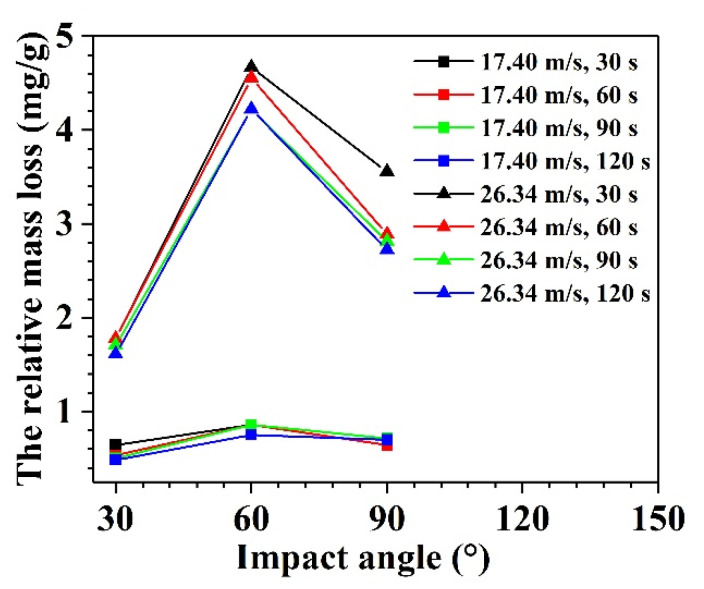
Relative mass loss of glass specimens with increase in impact angle.

**Figure 8 materials-14-00607-f008:**
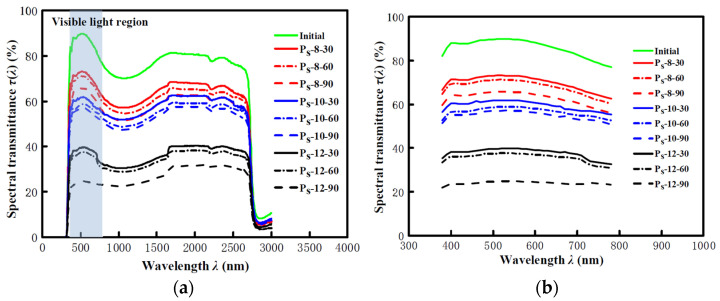
Spectral transmittance *τ* (*λ*) of the glass specimens without and with erosion for 5 s: (**a**) full spectral transmittance *τ* (*λ*) (300 nm ≤ *λ* ≤ 3000 nm) of glass specimens without and with erosion for 5 s and (**b**) visible spectral transmittance *τ* (*λ*) (380 nm ≤ *λ* ≤ 780 nm) of glass specimens without and with erosion for 5 s.

**Figure 9 materials-14-00607-f009:**
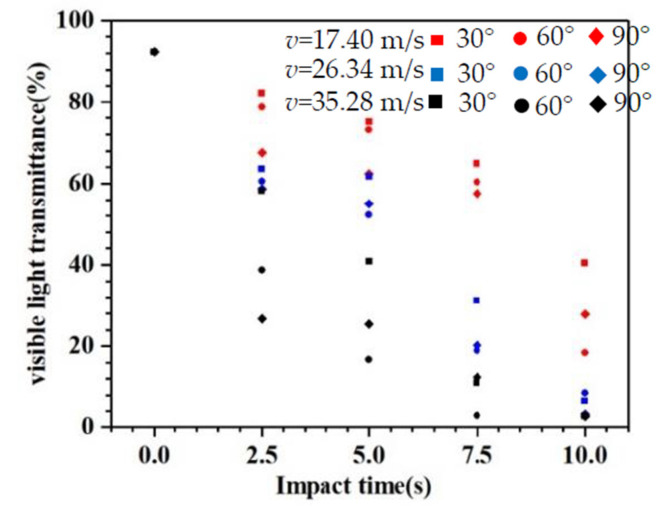
Visible light transmittance of the glass specimens *τ_v_*.

**Figure 10 materials-14-00607-f010:**
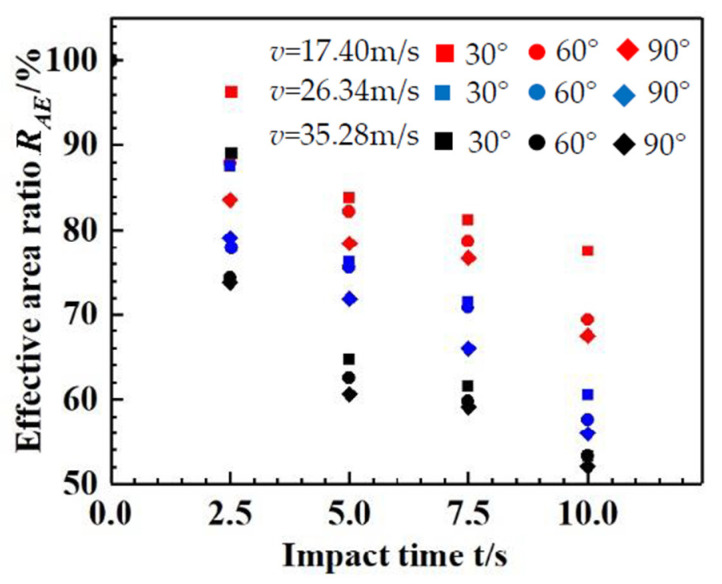
Relationship between the effective area ratio *R_AE_* of glass specimens and impact time *t*.

**Figure 11 materials-14-00607-f011:**
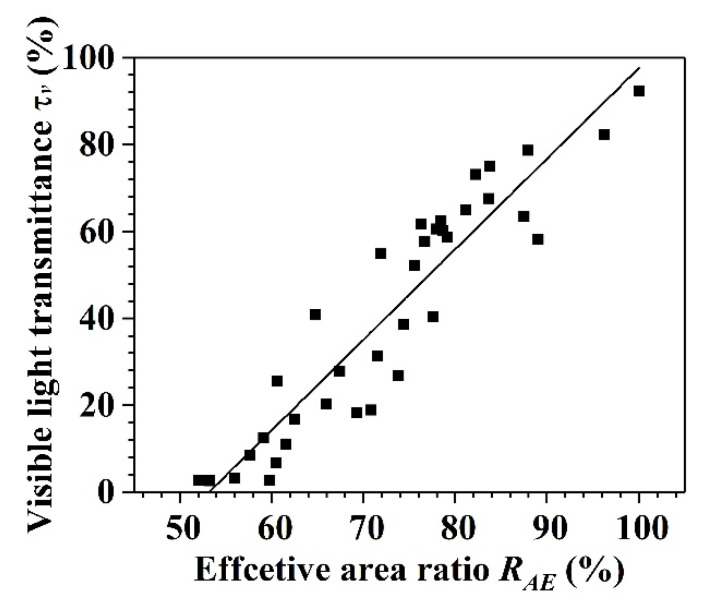
Relationship between visible light transmittance *τ_v_* and effective area ratio *R_AE_*_._

**Table 1 materials-14-00607-t001:** Basic information of glass.

Thickness *T*/mm	Gravity Density *γ_g_*/(kN/m^3^)	Initial Visible Transmittance *τ*_*v*0_/%	Design Strength *f_g_*/(N/mm^2^)	Modulus of Elasticity *E_g_*/(N/mm^2^)	Poisson Ratio *V*
6	25.6	92.34	28	0.72 × 10^5^	0.20

**Table 2 materials-14-00607-t002:** Specimens of the glass panels.

Specimen Type	Specimen Number	Abrasive Feed Rate *m* (g/s)	Impact Velocities *v* (m/s)	Impact Angles *a* (°)	Impact Time *t* (s)
P_a_	P_a_-1-30	15.5	17.40	30	0, 30, 60, 90, 120
	P_a_-1-60	15.5	17.40	60	
	P_a_-1-90	15.5	17.40	90	
	P_a_-2-30	15.0	26.34	30	
	P_a_-2-60	15.0	26.34	60	
	P_a_-2-90	15.0	26.34	90	
P_s_	P_s_-1-30	15.5	17.40	30	0, 2.5, 5, 7.5, 10
	P_s_-1-60	15.5	17.40	60	
	P_s_-1-90	15.5	17.40	90	
	P_s_-2-30	15.0	26.34	30	
	P_s_-2-60	15.0	26.34	60	
	P_s_-2-90	15.0	26.34	90	
	P_s_-3-30	14.8	35.28	30	
	P_s_-3-60	14.8	35.28	60	
	P_s_-3-90	14.8	35.28	90	

## Data Availability

Data is contained within this article. The data presented in this study are available. Anyone in need can use it or contact with me directly.

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
