# Peer review of "Windblown Sand-Induced Degradation of Glass Panels in Curtain Walls"

_materials, 2021, doi:10.3390/ma14030607_

Round 1

Reviewer 1 Report

The paper "Windblown Sand-induced Degradation of Glass Panels in Curtain Walls" presents a wind-blown sand experiment to explore the degradation law(s) of glass panels. Its content is compliant with the goal of Materials. I think it is innovative and never published. The format does not comply with the format of the Journal (.e.g bold letters, figures, tables). References format should be edited according to the journal’s style. Journals’ names must be in abbreviated format . The format for journal papers should be as: ‘Author 1, A.B.; Author 2, C.D. Title of the article. Abbreviated Journal Name Year, Volume, page range.’ For style of other kinds of documents, please check template document. Use N-dash for the page range, not hyphen. Include the digital object identifier (DOI) for all references where available.

Author Response

I hava edited my manscript for complying with the format of the Journal, including bold letters, figures, tables and references

Reviewer 2 Report

The title is clear.

The subject matter is within the scope of the journal.

This article contains new aspects, but the authors must underline the major findings of their work, and explain how yours results represents a progress. Please clearly explain the novelty.

The manuscript not adheres to the journal's standards.

The Abstract section must be improved. The Abstract should refer to the study findings, methodologies, discussion as well as conclusion. In this form the abstract is too generally.

The key words permit found article in the current registers or indexes, but must be put in alphabetical order.

In the introduction it is not clearly described the state of the art of the investigated problem. More references are necessary. The references from last years are necessary for demonstrated that this study is actual.

Please verify:

Hao [4] and Zhang [5] used the same method…

Reference [4] is not Hao, and for reference [5] is Zhang et al.,

For Figure 1 (2 in manuscript) the authors have accept for use?

The methods aren’t well described.

I don’t understand:

Line 97-102:

The Materials and Methods should be described with sufficient details to allow others to replicate and build on the published results. Please note that the publication of your manuscript implicates that you must make all materials, data, computer code, and protocols associated with the publication available to readers. Please disclose at the submission stage any restrictions on the availability of materials or information. New methods and protocols should be described in detail while well-established methods can be briefly described and appropriately cited.

Experimential program … please correct.

Please present precision of …. An electronic balance was used to weigh

Please present device for Visible Light Transmittance Ratio and Shading Coefficient Detector (manufacturer, city, country). For SEM presents same information.

First part from: 3.1. Damage mode… must in Introduction. In Results and discussion must be shown the results from this study.

Figure 6: Fig 6. Relative mass loss of ductile and brittle materials with increase of impact angle [10], is necessary? The authors have accept?

The figures don’t have a good quality.

Figure 1 missing from manuscript.

In Figures there are mistakes: for example in Fig. 8 (a) visil. Please verify x axe in Fig 11. Relationship between visible transmittance τv and effective..

Table 1 is twice in manuscript. Please correct the figures and table numbering.

In tables are presented necessary results.

The manuscript is relatively easy to understand by scientists form other area.

The sentence is not very clear: According to Eq. 4, for every 1% decrease in the effective area ratio, the visible light transmission ratio decreases by 2.085% when the effective area ratio ranges from 54.419 295 to 97.431.

The conclusion must be revised. The Conclusion must contain major finding of experimental study.

The guide of authors is not respected. Please verify all references and respect guide for authors.

Please complete references.

Please respect journal abbreviation.

The paper was written in standard, grammatically correct English, more corrections are necessary.

Please respect Guide for authors!

Author Response

1.This article contains new aspects, but the authors must underline the major findings of their work, and explain how yours results represents a progress. Please clearly explain the novelty.

A: our team got the tendency of the relative mass loss, the visible light transmittance and effective area ratio. the changes of relative mass loss is not in accordince with the tendency of typical brittle materials. the tendengcy of visible light transmittance and effective area ratio was not studied. 

2. The Abstract section must be improved.

A: I have revised my Abstract for reading easily. it can be founded from Abstract that the experiment program and our finds.

3.The Key words must be put in alphabetical order

A: completed

4.In the introduction it is not clearly described the state of the art of the investigated problem. More references are necessary. 

A: There is few study on the  investigation problem. Some references that did not  cover is not referential.

5.Reference [4] is not Hao, and for reference [5] is Zhang et al.

A: completed

6.For Figure 1 (2 in manuscript) the authors have accept for use?

I have obtained the license to use the Figure 1.

7.Line 97-102

It is a mistake.I have deleted it.

8.Experimential program … please correct.

A:I have corrected it

9.Please present precision of …. An electronic balance was used to weigh

A: completed

10.Please present device for Visible Light Transmittance Ratio and Shading Coefficient Detector (manufacturer, city, country). For SEM presents same information.

A: completed

11.First part from: 3.1. Damage mode… must in Introduction. In Results and discussion must be shown the results from this study.

A: In Result and discussion I have shown the result from this study.

12.The figures don’t have a good quality.

A: I have improved the quality of the figures.

13.Figure 1 missing from manuscript.

A: completed

14.Table 1 is twice in manuscript. Please correct the figures and table numbering.

A: completed

15.The sentence is not very clear: According to Eq. 4, for every 1% decrease in the effective area ratio, the visible light transmission ratio decreases by 2.085% when the effective area ratio ranges from 54.419 295 to 97.431.

A: completed.

16.The conclusion must be revised. 

A:completed

17.The guide of authors is not respected. Please verify all references and respect guide for authors.

A:completed

Reviewer 3 Report

The authors present a paper on the degradation processes of glass panels due to windblown sand. The work in general is interesting and well presented, however, there are still some doubts that should be clarified before the paper is considered for publication.

In the introduction, the authors present a series of works that were carried out on the same theme. However, it would be interesting, instead of just listing the papers, to present some of their most relevant conclusions. In this way, it would be possible to justify the need for the present study and the presentation of its level of innovation. What does this work add to what is already known on this topic? It is essential to frame this work (mainly its relevance) with the current state of knowledge on the topic.

Is the test methodology presented in section 2.3 framed by a normative document? Was it invented by the authors? How did the other studies do these tests?

I do not understand the content of the last paragraph of section 2.3.

Was the glass used in the present study conventional glass or did it have any special treatment?

What is the correlation between the results obtained in this work and the practical (real) cases of windblown sand-induced degradation? Are the results obtained here comparable with real cases?

Does the test method used require no calibration process so that it can effectively reflect the real cases of windblown sand-induced degradation?

What is the impact in terms of safety of the glass panels of the three modes damage found?

The conclusions cannot be just a summary of the results obtained! There must be a global critical analysis of the problem initially formulated (evidently based on the results obtained).

Author Response

1.In the introduction, the authors present a series of works that were carried out on the same theme. However, it would be interesting, instead of just listing the papers, to present some of their most relevant conclusions. In this way, it would be possible to justify the need for the present study and the presentation of its level of innovation. What does this work add to what is already known on this topic? It is essential to frame this work (mainly its relevance) with the current state of knowledge on the topic.

A:For this manscript, The conclusions of the relevant literature are not useful for our study. the conclusion we got is worth reading.

2.Is the test methodology presented in section 2.3 framed by a normative document? Was it invented by the authors? How did the other studies do these tests?

A: The test methodology is not presented in section 2.3 framed by a normative documen. the other studies do these test with similar way. our team modified the experimental method appropriately according to the experimental conditions

3.I do not understand the content of the last paragraph of section 2.3.

A:It is a mistake, and I have deleted it.

4.Was the glass used in the present study conventional glass or did it have any special treatment?

A:No, it is a common float glass.

5.What is the correlation between the results obtained in this work and the practical (real) cases of windblown sand-induced degradation? Are the results obtained here comparable with real cases?

A:This manscript mainly involves the experimental study on float glass panels. There is no study between experiment and real cases.

6.Does the test method used require no calibration process so that it can effectively reflect the real cases of windblown sand-induced degradation?

A:Yes.

7.What is the impact in terms of safety of the glass panels of the three modes damage found?

A:In the manscript, the value of the relative mass loss is used for the degree of safety of the glass panels. Therefore, the degradation of the reative mass loss is the impact in terms of safety of the glass panels of the three modes damage found.

8.The conclusions cannot be just a summary of the results obtained! There must be a global critical analysis of the problem initially formulated (evidently based on the results obtained).

A:completed.

Round 2

Reviewer 1 Report

The paper can be accepted.

At line 89, "2.3" should be moved at line 90.

Reviewer 2 Report

More corrections were made.

Reviewer 3 Report

The work has conditions to be considered for publication.